# Real-space observations of 60-nm skyrmion dynamics in an insulating magnet under low heat flow

Xiuzhen Yu [1✉], Fumitaka Kagawa [1,2], Shinichiro Seki[2,3], Masashi Kubota [1,5], Jan Masell [1], Fehmi S. Yasin [1], Kiyomi Nakajima[1], Masao Nakamura [1], Masashi Kawasaki[1,2], Naoto Nagaosa[1,2] & Yoshinori Tokura [1,2,4]

Thermal-current induced electron and spin dynamics in solids –dubbed "caloritronics"– have generated widespread interest in both fundamental physics and spintronics applications. Here, we examine the dynamics of nanometric topological spin textures, skyrmions driven by a temperature gradient $\nabla T$ or heat flow, that are evaluated through in-situ real-space observations in an insulating helimagnet $Cu_2OSeO_3$. We observe increases of the skyrmion velocity and the Hall angle with increasing $\nabla T$ above a critical value of ~ 13 mK/mm, which is two orders of magnitude lower than the $\nabla T$ required to drive ferromagnetic domain walls. A comparable magnitude of $\nabla T$ is also observed to move the domain walls between a skyrmion domain and the non-topological conical-spin domain from cold to hot regions. Our results demonstrate the efficient manipulation of skyrmions by temperature gradients, a promising step towards energy-efficient "green" spintronics.

[1] RIKEN Center for Emergent Matter Science (CEMS), Wako, Japan. [2] Department of Applied Physics, University of Tokyo, Tokyo, Japan. [3] Institute of Engineering Innovation, University of Tokyo, Tokyo, Japan. [4] Tokyo College, University of Tokyo, Tokyo, Japan. [5] Present address: Technology and Business Development Unit, Murata Manufacturing Co., Ltd., Kyoto, Japan. ✉email: yu_x@riken.jp

Control and manipulation of electron spins are indispensable for the development of low energy-consumption electronic devices. Among them, electron transport phenomena induced by interaction with heat flow as known as thermal current have attracted significant attention since they promise to recycle the wasted heat facilitating green information technology. In the field of spin-caloritronics[1,2], a thermal current is associated with the spin current which induces emergent phenomena of the spin-charge coupled dynamics with related entropy[1], to name a few, spin Nernst effect[3], spin-dependent Peltier effect[4], spin Seebeck effect[5], and thermal Hall effect[6]. These thermal current-induced effects are mediated by the collective dynamics of spin textures, such as magnetic twists (domain walls) or vortex-like spin textures, coupled with a spin current $J_S$ converted from the thermal current[7]. Due to the conservation of total angular momentum in the system, the thermal current can in turn exert a torque on the spin texture that leads to temperature-gradient-induced spin texture dynamics[8,9]. In analogy to current-driven spin texture motions[10,11], temperature-gradient-induced magnetization dynamics have been reported in metallic systems via spin-transfer torque on magnetic domain walls[12]. Recent research reveals the role of temperature in the dynamics of topological spin textures (skyrmions):[13] the nonlinearly increasing Hall effect as a function of the skyrmion velocity in the low-drive regime close to the depinning threshold; the current-induced the deformation of skyrmion shape from a regular circle to an irregular one in the high-drive regime. A thermal gradient in metallic systems is also expected to cause skyrmion dynamics since the thermal current can induce both a magnon current and an electric current (thermopower). The electric current then exerts a force on the skyrmions via spin-transfer torque or spin-orbit torque, resulting in a complicated interaction between spin textures and heat current. These complex thermal effects have been demonstrated by recent work on the temperature gradient-driven skyrmion diffusion in a metallic multi-layer film[14].

In magnetically ordered insulators, however, charge currents are absent. Still, a temperature gradient can drive magnetic textures in insulating ferromagnets (FM)[15,16] via the transfer of spin angular momentum of the thermally induced magnon current, similar to the conduction electron spin-transfer torque mechanism in metallic FM. The possibility of inducing magnon currents and relatively smaller Gilbert damping constant in insulating FM is attracting considerable attention for energy-efficient manipulations of spin textures.

Compared with topologically trivial spin textures like FM domain wall, vortex-like topological spin textures, e.g., magnetic skyrmions, are expected to move with lower temperature gradients $\nabla T$ due to their topological nature, which is advantageous for low energy consumption in electronic devices[17–22]. In particular, skyrmions emerging in insulating materials are attractive for applications that require low energy dissipation. The magnon current $j_x$ in the lateral direction produces a skyrmion velocity as described by[23]

$$v_x = v_x^M - v_x^B \qquad (1)$$

$$v_y = 2\alpha\eta Q v_x^M \qquad (2)$$

where $v_x^M = \gamma J a^2 j_x$ is the velocity due to the magnon current, and

$$v_x^B = \left(\frac{\gamma}{\pi Q^2}\right)\alpha\eta a^2 k_B \nabla T \qquad (3)$$

is that due to the Brownian motion. Here $a$, $J$, $\alpha$, and $\gamma$ are the lattice constant, the exchange coupling constant, the Gilbert damping constant, and the gyromagnetic ratio, and $k_B$, $\nabla T$, and $\eta$ are the Boltzmann constant, the temperature gradient, and a

constant of order unity, respectively. The $Q$ is the skyrmion number (integer topological charge), as defined by

$$Q = \frac{1}{4\pi}\int M \cdot \left(\frac{\partial M}{\partial x} \times \frac{\partial M}{\partial y}\right) dx dy. \qquad (4)$$

Here $M$ is the unit vector of local magnetization in the related materials. When the Gilbert damping constant $\alpha$ is small, the Brownian motion can be neglected compared to the magnon-driven skyrmion motion[24].

Among insulating magnets, the helimagnet $Cu_2OSeO_3$ hosting 60-nm skyrmions is a good target material[25] to manipulate topological spin textures with an electric field[26–28] owing to its intrinsic magnetoelectric coupling. In addition, thermally driven unidirectional rotation of a skyrmion lattice (SkL) has been demonstrated in a thin plate of $Cu_2OSeO_3$ under the presence of radial thermal current[29]. The next step in exploring such thermally driven skyrmion dynamics is to understand the drift motion of isolated skyrmions or skyrmion clusters in a spin-polarized background under a linear $\nabla T$, and to quantitatively evaluate the threshold $\nabla T$ for driving skyrmions. Similar to the thermal control of FM domain walls in an insulator, skyrmion dynamics with a magnon current have been numerically predicted in several studies[23,30–33]. The results indicate that skyrmions move from cold to hot regions via a magnonic spin-transfer torque, in the opposite direction of the magnon flow. However, these predictions have yet to be observed.

Here we apply $\nabla T$ to a thin plate of $Cu_2OSeO_3$ crystal equipped with a heater and thermometers. We use Lorentz transmission electron microscopy (TEM) to perform in situ real-space observations of the proliferation and drift motion of 60-nm skyrmions and evaluate their dynamics under the low $\nabla T$.

## Results

**Thermally driven skyrmion motions in a designed microdevice composed of an insulating magnet.** $Cu_2OSeO_3$ has a chiral-lattice structure as shown in Fig. 1a. The hexagonal SkL can be generated under the specific fields and temperatures, as exemplified by an over-focus Lorentz TEM image taken at 20 K under a normal magnetic field of 60 mT (Fig. 1b). To investigate the skyrmion dynamics under temperature gradients, we have prepared a $Cu_2OSeO_3$ thin plate with a heater (H) and thermometers (R1 and R2) composed of the meander Pt pattern (the size: $l$ (length in total) = 2.34 mm, $w$ (width) = 2.1 μm, and $t$ (thickness) = 25 nm) on the top surface of the plate having a partially thinner region for electron beam transmission, as shown in Fig. 1c, d; the top Fig.1c and cross-section Fig.1d views, respectively. Figure. 1e is a schematic drawing of skyrmion flows under $\nabla T$ in the lateral direction, illustrating that a skyrmion moves from a cold region to hot region against the heat flow and exhibits simultaneously Hall motion.

**Realizations of various magnetic configurations including skyrmions in the microdevice.** Figure. 2 represents the externally applied magnetic field ($B$)–temperature ($T$) phase diagram (Fig. 2a) and corresponding magnetic configurations of the 100-nm thin $Cu_2OSeO_3$ (Fig. 2b–g) measured by systematic Lorentz TEM observations at various $T$ and $B$ applied normally to the thin plate. Multiple domains of the in-plane helices (stripes in Fig. 2b) appear in the initial state, as we increase field $B$, mixed spin textures emerge, such as coexisting helices (H) and SkL (Fig. 2c, d), and SkL coexistent with a conical (C) domain (Fig. 2f). The nearly perfect SkL can only be generated under proper $T$ and $B$ conditions (Fig. 2e), and a FM state (spin-polarized state) dominates under a higher external field (Fig. 2g). (Note here that Lorentz TEM gives monotonic contrast for the vertical C structure and an FM state.) In the following, we have chosen mixed states consisting

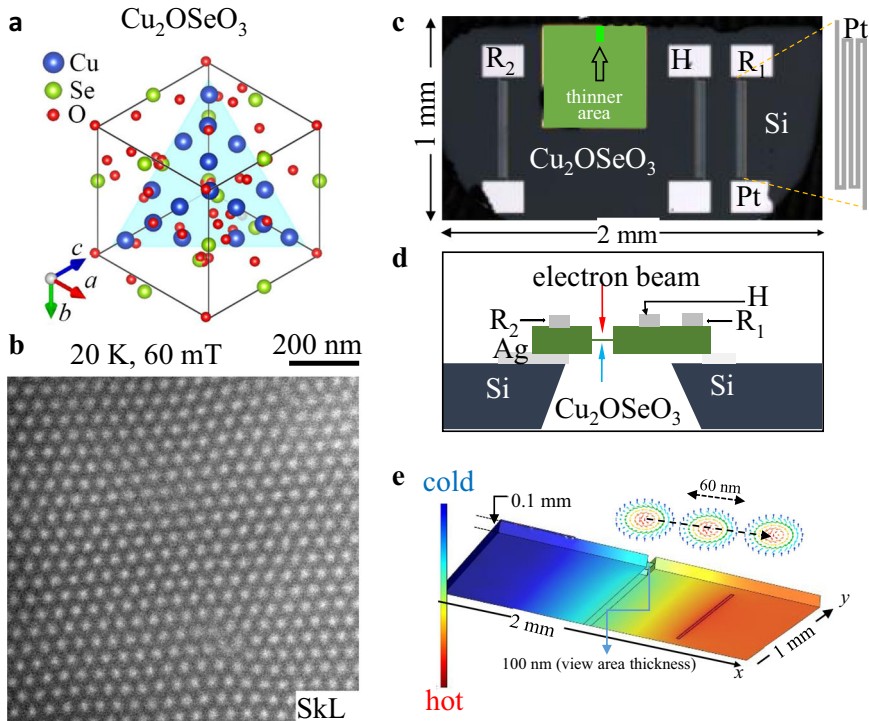

**Fig. 1 Thermally driven magnetic skyrmion motion in an insulating magnet Cu₂OSeO₃ with chiral-lattice structure. a** Schematic crystal structure of the Cu₂OSeO₃. **b** A skyrmion (white dot-like contrasts) lattice (SkL) observed by Lorentz transmission electron microscopy (TEM) in a (111) Cu₂OSeO₃ thin plate under a normal 60 mT field at 20 K. **c**, **d** Device configurations (**c** topography of the device; **d** schematic of the device cross-section) for imaging skyrmion dynamics with heat flows. **e** Schematics of skyrmion flows in the thin plate with temperature gradients ($\nabla T$). Dashed arrows indicate the anticipated trace of a skyrmion.

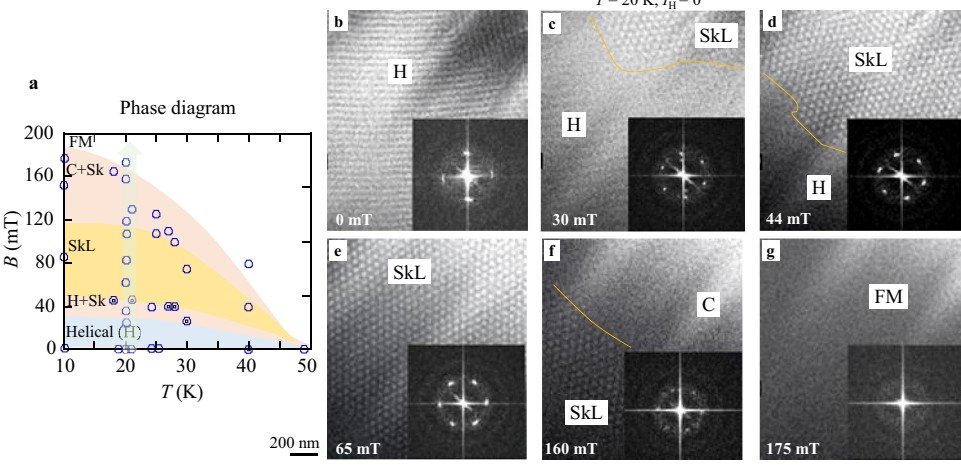

**Fig. 2 The temperature (*T*) - magnetic field (*B*) phase diagram observed in Cu₂OSeO₃ by Lorentz TEM. a** The *T-B* phase diagram of magnetic structure in the (111) 100-nm thick Cu₂OSeO₃ thin plate. Circles specify the *T* and *B* conditions. H, C, and FM stand for the helical, conical, and field-aligned ferromagnetic structures, respectively. **b**–**g** Lorentz TEM images and their fast Fourier transforms (FFTs) were observed at 20 K with an increasing magnetic field.

of skyrmions and C or FM domains to probe the temperature-gradient effect on skyrmions.

**Skyrmion dynamics in the microdevice under low heat flow.** Figure. 3a, b show two isolated islands of skyrmions observed before (Fig. 3a) and during (Fig. 3b) a 10-μA-current excitation on the heater at an indicated elapsed time (0.8 s), respectively. The center position of the skyrmion island does not change significantly in spite of the change of orientations (indicated by yellow and orange dashed lines in Fig. 3a, b, respectively) of its

hexagonal lattice during the current excitation. Figure. 3c, d show two snapshots of over-focus Lorentz TEM images for a mixed state with a skyrmion domain and possible vertical- C domain (monotonic contrast in the images), observed at a static state (Fig. 3c) and during (Fig. 3d) a 50-μA-current excitation at an elapsed time of 1.16 s, respectively. The facets of the hexagonal lattice of skyrmions as the domain walls between skyrmion domain and C domain, indicated by yellow dashed lines in Fig. 3c, drift from the cold region to the hot region (right side), accompanying a constriction of the C domain and possible

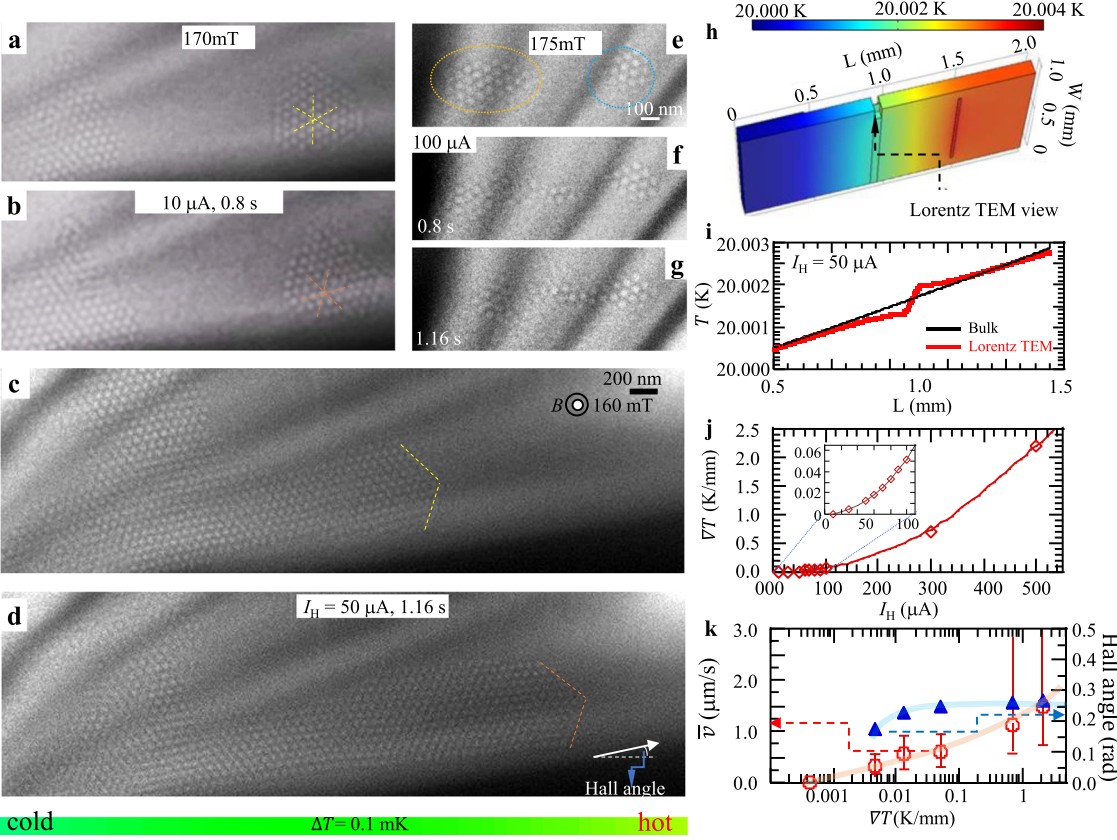

**Fig. 3 Heat flow-driven skyrmion motion in a Cu₂OSeO₃ thin plate. a, b** Lorentz TEM images observed before (**a**) and during (**b**) a 10 μA current flowing through the heater. **c** Skyrmion (white dot) domain coexistent with a vertical C domain (monotonic contrast) observed under a normal field of 160 mT at 20 K in the (111) thin Cu₂OSeO₃. The boundaries between the skyrmion domain and C domain are signed by yellow dashed lines. **d** The domain boundaries between skyrmions and C domain drift from the lower left to the upper right (indicated by orange dashed lines) when a 50 μA current flows through the heater (H) set on the right side of the device (Fig. 1c, d). **e–g** The left skyrmion island (encircled by a dotted yellow line) flows towards the right one (encircled by dotted blue line) with 100 μA current flow. **h** $T$-map of the thin Cu₂OSeO₃ during a 50 μA current flow. Color bar indicates the $T$-scale. **i** Line profiles of $\nabla T$ in the Lorentz TEM view area (the red line) and in the bulky Cu₂OSeO₃ (thicker regions, the black line) at $I_H = 50$ μA. **j** Calculated $\nabla T$ versus $I_H$ in the Lorentz TEM view area. The inset is an enlargement of the $\nabla T$ profile at a range of $I_H$ from 0 to100 μA. **k** Variation of the averaged velocity $\overline{v}$ (red circles) of the domain wall (the boundary between skyrmion domain and C domain) and Hall angle (blue triangular) of the front skyrmion at the boundary with an increase of $\nabla T$, observed while holding a constant field of 160 mT. The $\overline{v}$ is a ratio of the total drift distance to the duration (the duration is 1.36 s for the heater current $I_H = 0.3, 0.5$ mA, while it is 1.68 s for $I_H = 0.3, 0.05, 0.1$ mA) of skyrmion motions. The error bars represent the maximal and minimal averaged velocities deduced from the locally-averaged values approximately over the shorter time period of 0.2–0.6 s, as detailed in Supplementary Fig. 1. The pink and blue lines are eye guides for the changes of the velocity and Hall angle, respectively.

proliferation of metastable skyrmions or flows of skyrmions from the thicker region on the left outside to the right inside of the Lorentz TEM view. The latter is difficult to confirm due to electron wave amplitude loss in the thicker sample region (no TEM contrast). When the heater current is turned off, the domain walls almost come back to the initial state (Fig. 3c), accompanying the replacements of metastable skyrmions by C domain or the movements of skyrmions from the view area to the left outside of the view, indicating reversible skyrmion dynamics with the heat flow. The reason for the reversible skyrmion dynamics with heat flow shown here is not clear at present; most likely are tiny residual strains in the sample, which determine the skyrmion distribution at the equilibrium state under zero heat flow. When the heat current is switched off, the skyrmions can either move back to this energetically preferred position, or other skyrmions can be attracted by this spot, or new skyrmions can be nucleated. The precise mechanism is, however, not traceable within the time resolution of our experiments. The abovementioned observations have demonstrated that skyrmion drift motion is nonzero as a function of heat flow (see also Supplementary Fig. 1 and an in situ Lorentz TEM movie, Supplementary Movie 1 in Supplementary

Information (SI)) only for the heater currents $I_H \geq 50$ μA. Upon further increase of the heater current up to 100 μA, we clearly discern the skyrmion translational motion, as shown in Fig. 3e–g. At the initial state, two isolated skyrmion islands encircled by yellow and blue dashed lines, respectively, were stabilized in different regions of the device (Fig. 3e). When we excite the heater with a 100 μA current, the left island starts to move toward the right one (Fig. 3f), and finally unites with the right island to flow towards the hot region (right side) of the device (Fig. 3g and Supplementary Movie 2).

According to the abovementioned real-space observations, sky-rmions in Cu₂OSeO₃ can be driven by a low heat power, P ($I_H^2 \times R$) ~ 10 μW ($I_H = 50$ μA, $R = 3800$ Ω). Such small heat generation does not induce appreciable changes of resistances of two thermometers $R_1$ and $R_2$ (see Supplementary Fig. 2) and hence can hardly characterize the $\nabla T$ in Lorentz TEM view area by resistance measurements. Therefore, to evaluate $\nabla T$ in the Lorentz TEM view area with varying $I_H$ in the experimental setup here, we have simulated the $\nabla T$ in the view area as well as the thicker bulky part of the Cu₂OSeO₃ in terms of the finite element method (see "Methods" and "SI" for details). Figure. 3h shows the simulated temperature

map of the device under 50 μA excitation on the heater; Fig. 3i shows related temperature gradients in the Lorentz TEM view area (red line) and the thicker bulky part (black line) of the $Cu_2OSeO_3$ plate, respectively. It indicates that, in the present parameter set (for details, see "Methods"), the $\nabla T$ is ~ 13 mK/mm in the thinner Lorentz TEM view area, several times larger than that (~ 2 mK/mm) in the thicker region of the $Cu_2OSeO_3$ plate under 50 μA excitation on the heater. Figure. 3j represents the $\nabla T$ profile as a function of the heater current $I_H$. To deduce skyrmion dynamics with varying $\nabla T$, the in situ Lorentz TEM data were obtained with a 40 ms exposure time (recording speed of 25 fps (frames per second)), and 160 mT magnetic field applied normally to the $Cu_2OSeO_3$ plate. Using the calculated values of $\nabla T$ and the systematic in situ Lorentz TEM observations with varying $I_H$, we plotted the averaged velocity ($\overline{v}$)(red circle symbols; defined by a ratio of the total drift distance to the duration) (the duration is 1.36 s for the heater current $I_H = 0.3$, 0.5 mA, while it is 1.68 s for $I_H = 0.3$, 0.05, 0.1 mA) of skyrmion motion; the error bars represent the maximal and minimal velocities deduced from the locally averaged values approximately over the shorter period of 0.2–0.6 s, as detailed in the Supplementary Fig. 1, and Hall angle (blue triangular symbols, defined as an angle between the heat flow and skyrmion flow directions (Fig. 3d)) of the skyrmions with respect to $\nabla T$, as shown in Fig. 3k; the results were obtained by tracking the cusp point of the skyrmion domain (Fig. 3c, d). The Hall motion in the upward direction and the sliding motion in the opposite direction of heat flow can be clearly discerned. Both the Hall angle and $\overline{v}$ increase nonlinearly with increasing $\nabla T$, and the $\overline{v}$ is ~ 0.6 μm/s at $\nabla T$ ~ 13 mK/mm. Note that the Hall angle increases nonlinearly with the $\overline{v}$, as shown in Supplementary Fig. 4. The dependence of the Hall angle on the velocity of skyrmions drivenby the electric current has been observed experimentally[34,35]. Also the role of disorder on the skyrmion Hall effect has been studiedtheoretically[36,37]. The present results are the first experimental observation of heat-driven skyrmion motion in the insulating magnet.

## Discussion

The above-mentioned experimental results are roughly in accord with our theoretical estimations. Nominally, the magnon current is given by[23,32]

$$j_x = \int v_x(k)\delta f(k)\frac{d^d k}{(2\pi)^d} \tag{5}$$

where $v_x(k) = \frac{\partial \epsilon(k)}{\hbar \partial k_x}$ is the velocity of a magnon with momentum $k$, and $\delta f(k) = \tau v_x \frac{\partial f_0}{\partial \epsilon}\frac{\nabla T}{T}$ is the nonequilibrium distribution correction. Here, $\tau$ is the magnon lifetme, $\epsilon$ its energy, $f_0 = \left(\exp\left(\frac{\epsilon}{k_B T}\right) - 1\right)^{-1}$ the Bose–Einstein distribution, $d$ the dimensionality of the material system, and $T$ the temperature. For a temperature $T = 20$ K and the Gilbert damping of the order $\alpha \sim 6 \times 10^{-3}$ (ref. [38]), we roughly estimate a magnon current of the order

$$j_x \sim \sim -9 \times 10^{20}\nabla T(K^{-1}s^{-1}m^{-1}) \tag{6}$$

skyrmions (with $\nabla T$ in the unit of K/m) which results in a magnon-induced velocity of

$$v \approx 0.16 \times 10^{-6}\nabla T\left(\frac{m^2}{s \cdot K}\right).$$

When $\nabla T \approx 50$ mK/mm, the velocity of a skyrmion is estimated as $v = 8$ μm/s, which is one order of magnitude larger than the value obtained in the present experimental setup.

In summary, we have demonstrated the dynamics of small-size (60 nm) skyrmions driven by a temperature gradient ($\nabla T$) across thin plates composed of insulating $Cu_2OSeO_3$ using real-space Lorentz TEM imaging. We observed the drift of the domain wall

between the non-topological magnetic structure and a skyrmion domain, as well as the propagation of skyrmions under the low $\nabla T$. We found that the velocity and Hall angle of skyrmions increases with increasing $\nabla T$ in the low thermal-current regime and observed a threshold value of $\nabla T$ ~ 13 mK/mm which is two orders of magnitude smaller than that needed to drive conventional FM domain walls.

## Methods

**Sample preparation and Lorentz TEM observations**. The heater and thermo-meters made of meander Pt pads (Fig. 1c) were put on the top of a $Cu_2OSeO_3$ plate by depositing Pt film on the surface of $Cu_2OSeO_3$ and then by patterning and etching Pt film using electron lithography. To perform in situ Lorentz TEM observations for the $Cu_2OSeO_3$ with heat flows, the plate was partially thinned down to 100-nm thick by a focused ion beam system (NB5000, Hitachi), and then was fixed on the Si-substrate which was settled on the TEM sample holder with electrodes.

All the real-space observations of skyrmions in the plate were carried out at the Lorentz TEM mode after zero-field cooling by using a multifunctional TEM (JEM-2800), which is equipped with a fast-imaging system (Gatan, OneView) and a single-tilt liquid He sample holder (Gatan, HLTST) with ten electric terminals connected to three power sources (Keithley, 2612B). In the defocused Lorentz TEM images, skyrmions appear as bright/dark dots in the overfocused/under focused Lorentz TEM images, reflecting their nature of clockwise spin helicity in the $Cu_2OSeO_3$ (ref. [25]).

**Simulations of temperature maps in devices under heat flows**. In the finite element simulations of the temperature gradient of the present $Cu_2OSeO_3$, we used a commercially available software, COMSOL Multiphysics (COMSOL Inc.). To mimic the real device structure (Fig. 1c, d), we constructed a model: the bottom of one side of the thin plate is contacted with a "cold bath" ($T = 20$ K) and that of the opposite side is floated from the bath. The thermal conductivity and specific heat of $Cu_2OSeO_3$ are set to be typical 20 K values according to the literature[39]; they are 40 W/m K and 4 J/mol K, respectively. The resistance heater is put on the surface of the floating side, and its resistance value is set to be 3800 Ω (the measured value). The spatial profile of the temperature is thus calculated for the case that DC current of 10–500 μA is applied to the resistance heater.

## Data availability

The data that support the findings of this study are available from the corresponding author upon reasonable request.

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

## Acknowledgements

The authors thank Sadamichi Maekawa, Max Hirschberger, and Naoki Ogawa for fruitful discussions. This work was supported in part by Grants-In-Aid for Scientific Research (A) (Grant No. 19H00660) from Japan Society for the Promotion of Science (JSPS), JSPS program (project No. 19F19815) and the Alexander von Humboldt Foundation, and Japan Science and Technology Agency (JST) CREST programs (Grant Number JPMJCR1874, Grant Number JPMJCR20T1), Japan.

## Author contributions

X.Y., M.Kawasaki, N.N., and Y.T. conceived the project. X.Y. performed Lorentz TEM observations, analyzed the experimental data, and wrote the manuscript with Y.T. J.M. and N.N. contributed to the theoretical analysis. S.S. synthesized the bulky $Cu_2OSeO_3$. X.Y., F.K., M.K, M.N., F.S.Y., and K.N. prepared devices and characterized temperature gradients in devices. All authors discussed the data and commented on the manuscript.

## Competing interests

The authors declare no competing interests.
