## [Peer Review File · Nature Communications]

Reviewers' Comments:

Reviewer #1:

Remarks to the Author:

Report for

Real-space observations of 60-nm skyrmion dynamics in an insulating magnet under low 1 heat flow

This paper reports on imaging experiments for the motion of skyrmions in insulating magnetic systems under a thermal gradient. There has been many studies of skyrmion motion under currents but far less work has been done on the motion of skyrmions due to heat flow. Skyrmions in insulators have numerous properties that make them distinct from skyrmions in non-insulators, such as a strong non-adiabatic component to their dynamics and their size scale. They also offer unique opportunities for applications that are distinct from those for skyrmions driven with currents. For skyrmions driven with a current there have been several observations of the skyrmion Hall effect and this paper is the first observation that I am aware of demonstrating the skyrmion Hall angle under a thermal gradient. In addition to the skyrmion Hall angle, the authors also show that the Hall angle increases with skyrmion velocity, which is again an effect observed in current driven systems but not previously found for thermally driven systems. Another aspect of the paper is that the temperature gradient required to move the skyrmions is two orders of magnitude less than that for domain walls. The results for the thermally generated motion are also in agreement with previous theoretical studies confirming those results. The paper is clear and the results appear correct and should generate additional studies in skyrmion dynamics in the single and lattice limit under thermal gradients.

I think the paper is very nice and should be published and I have only one point for the authors to comment on.

The observation of the dependence Hall angle increasing with thermal gradient is very interesting. For current driven systems this effect was predicted to occur due to the combination of the Magnus force and pinning effect.

"Collective transport properties of driven Skyrmions with random disorder",
C. Reichhardt, D. Ray, and C.J. Olson Reichhardt
Phys. Rev. Lett. 114 , 217202 (2015).

"Noise fluctuations and drive dependence on the skyrmion Hall effect in disordered systems",
C. Reichhardt and C.J. Olson Reichhardt
New J. Phys 18 095005 (2016).

Such an effect has been seen in several experiments such as

Diameter-independent skyrmion Hall angle observed in chiral magnetic multilayers
Katharina Zeissler, Simone Finizio, Craig Barton, Alexandra J. Huxtable, Jamie Massey, Jörg Raabe, Alexandr V. Sadovnikov, Sergey A. Nikitov, Richard Brearton, Thorsten Hesjedal, Gerit van der Laan, Mark C. Rosamond, Edmund H. Linfield, Gavin Burnell & Christopher H. Marrows
Nature Communications volume 11, Article number: 428 (2020).

Other works have suggested that the effect in the current driven systems arises due to the current changing the shape of the skyrmion.

The role of temperature and drive current in skyrmion dynamics
Kai Litzius, Jonathan Leliaert, Pedram Bassirian, Davi Rodrigues, Sascha Kromin, Ivan Lemesh, Jakub Zazvorka, Kyu-Joon Lee, Jeroen Mulkers, Nico Kerber, Daniel Heinze, Niklas Keil, Robert M. Reeve, Markus Weigand, Bartel Van Waeyenberge, Gisela Schütz, Karin Everschor-Sitte, Geoffrey S. D. Beach & Mathias Kläui
Nature Electronics volume 3, pages30–36(2020)

The results in the manuscript under review show the velocity dependence of the skyrmion Hall angle occurs even without an applied current, which suggests that current induced shape changes of the

moving skyrmions are not key to producing the skyrmion Hall effect. The authors should mention the possible role of pinning on determining the Hall angle and for example if the authors see what appears to be a threshold, is this due to pinning or to some other effect?

Reviewer #2:

Remarks to the Author:

Review of "Real-space observations of 60-nm skyrmion dynamics in an insulating magnet under low heat flow" by Yu, et al. submitted to Nature Communications

The manuscript "Real-space observations of 60-nm skyrmion dynamics in an insulating magnet under low heat flow" presents the direct observation of translational and lateral motion of skyrmions driven by a temperature gradient in the insulating helimagnet Cu₂OSeO₃ using Lorentz transmission electron microscopy (Lorentz TEM). The temperature gradient is established by applying an electric current into the on-chip heater made of meander Pt stripes using electron lithography on the surface of Cu₂OSeO₃, and the 100µm-thick Cu₂OSeO₃ thin plate was partially thinned down to 100nm thick using a focused ion beam system so as to perform the in-situ Lorentz TEM observation. By changing the environment temperature T and normally applied magnetic field B, a phase diagram of magnetic structures in Cu₂OSeO₃ is established through analyzing the obtained Lorentz TEM images as well as the corresponding fast Fourier transforms (FFTs) results. Then the authors applied different electric current into the heater and estimated their effects on the motion of skyrmions, during which a rather small current (10µA) can only change the orientation of skyrmions lattice, while obvious lateral and translational motion of skyrmion lattice can be witnessed when the current is higher (50µA and 100µA, respectively). The temperature gradient generated by electric current is quantitatively characterized using COMSOL simulation, and the temperature gradient needed in experiment for the velocity of skyrmion lattice to reach 0.1mm/s is of the same order with the theoretical estimation.

Considering the completeness of the available data, and the novelty presented in this manuscript, I failed to recognize its publication in Nature Communications. Below are my specific comments:

1. The crystal structure, Figure 1A, the formation of skyrmions in this materials, Figure 1B, the phase diagram, Figure 2, for the magnetic structures of Cu₂OSeO₃ have already been published in many different articles [1–3]. The results produced by authors present subtle differences from those published results.
2. Thus, this leaves only the schematic illustration in Figures 1c-1e, and Figure 3 as meaningful data in this manuscript. The discussion on these parts is, however, very limited. It is not clear how COMSOL simulation is being done, what are the parameters used for simulation? The identification of skyrmion motion, together with the device design are highly questionable. I am also wondering what is the purpose of mentioning 60 nm? Are there existing bigger skyrmions in Cu₂OSeO₃?
3. I am wondering why authors made their device out of a big thick plate? I am expecting to see more clean physics out of a single thin stripe, without complication from a big surrounding thick area.
4. Under such a small temperature gradient, it is surprising to see the "motion" of skyrmions. While, I can see the propagation of skyrmions from left towards the right regimes, in the presence of temperature gradient. I failed to see a more comprehensive experimental investigation of the reversible skyrmion motion. The phase diagram suggests skyrmions can exist in a wide temperature range, I am thus wondering what limits the available amplitude of temperature gradients?
5. I am not quite sure if it makes sense to talk about the skyrmion Hall angle in their experiment. After all, only skyrmions in particular region are movable. Their paths are thus confined/forced to that particular region (from bottom left to the up right), which in principle could give rise to a false estimation of skyrmion Hall effect, since their path is predetermined from the bottom left to the up right. Authors should reverse the current direction, and the polarity of magnetic field orientation, to see direction of skyrmion Hall motion.

6. As I am concerned, the authors modeled the whole Cu_2OSeO_3 thin plate and perform the simulation using COMSOL in order to characterize the temperature profile of the sample, during which the very distant edge from the heater of the plate is set to be 20K directly, while the temperature elsewhere is calculated accordingly. I am not quite sure if this is a proper way to obtain the precise temperature, since the Cu_2OSeO_3 thin plate actually gets contacted with the "real" cold bath (the sample holder) through Si-substrate. It seems to me more convincing and reliable to simulate the whole device including Cu_2OSeO_3 thin plate and the Si-substrate, as the temperature profile is crucial for the subsequent analysis. Or can the author verify their simulation about temperature?

7. Another major concern is the dragging of skyrmions back to the initial state after turning off the heater current is tuned off in Fig. 3c. What are the reasons underlying this observation? Does this mean that the sample is not uniform and the skyrmions are likely to stay in specific areas? Will the nonuniformity plays important roles in the motion of skyrmions as well (this question is similar to 5)

8. In Fig. 3e-g, why does the left island marked with yellow circle move and unite with the right island rather than moving simultaneously with the right island, keeping the distance between them unchanged, if two islands can both be driven by the temperature gradient?

9. It should mention here that the interaction between skyrmions and heat can be very complex, and depend on the detailed parameters, many different physics and direction of motion can be observed. For examples, a motion of skyrmions from hot towards cold regimes in metallic multilayer has been reported. This suggests a more comprehensive theoretical discussion, based on the available experimental parameters in their material system should be adequately discussed.

10. Many early (mostly theoretical) discussions on the interaction between skyrmions and heat did not get noticed. The cited references on spin caloritronics are too general and not specific for the present manuscript.

In summary, authors observed the motion of skyrmion from cold towards hot regime. Their experimental observations demand further comprehensive investigations. Whether the motion of skyrmion is a result of magnonic spin torque remains further justification. The observation of skyrmion Hall angle is not adequately discussed, which probably occurs as a result of the uniformity of their sample. The theoretical discussion is rather vague, the simulation requires a more careful revisit. This manuscript requires a substantial experimental inputs, and thus I am not recommend its publication in in Nature Communications.

Response to Reviewers' Comments

In the following, we reply to the reviewers' comments one by one. For better visibility, we cite the reviewers' comments in *blue*.

Response to Reviewer #1

1) This paper reports on imaging experiments for the motion of skyrmions in insulating magnetic systems under a thermal gradient. There have been many studies of skyrmion motion under currents but far less work has been done on the motion of skyrmions due to heat flow. Skyrmions in insulators have numerous properties that make them distinct from skyrmions in non-insulators, such as a strong non-adiabatic component to their dynamics and their size scale. They also offer unique opportunities for applications that are distinct from those for skyrmions driven with currents. For skyrmions driven with a current there have been several observations of the skyrmion Hall effect and this paper is the first observation that I am aware of demonstrating the skyrmion Hall angle under a thermal gradient. In addition to the skyrmion Hall angle, the authors also show that the Hall angle increases with skyrmion velocity, which is again an effect observed in current driven systems but not previously found for thermally driven systems. Another aspect of the paper is that the temperature gradient required to move the skyrmions is two orders of magnitude less than that for domain walls. The results for the thermally generated motion are also in agreement with previous theoretical studies confirming those results. The paper is clear and the results appear correct and should generate additional studies in skyrmion dynamics in the single and lattice limit under thermal gradients. I think the paper is very nice and should be published.

We thank the reviewer for evaluations of our present work and his/her recommendation to publish our manuscript in Nature Communications.

2) I have only one point for the authors to comment on. The observation of the dependence Hall angle increasing with thermal gradient is very interesting. For current driven systems this effect was predicted to occur due to the combination of the Magnus force and pinning effect. "Collective transport properties of driven Skyrmions with random disorder", C. Reichhardt, D. Ray, and C.J. Olson Reichhardt Phys. Rev. Lett. 114, 217202 (2015). "Noise fluctuations and drive dependence on the skyrmion Hall effect in disordered systems", C. Reichhardt and C.J. Olson Reichhardt New J. Phys 18 095005 (2016). Such an effect has been seen in several experiments such as Diameter-independent skyrmion Hall angle observed in chiral magnetic multilayers Katharina Zeissler, Simone Finizio, Craig Barton, Alexandra J. Huxtable, Jamie Massey, Jörg Raabe, Alexandr

V. Sadovnikov, Sergey A. Nikitov, Richard Brearton, Thorsten Hesjedal, Gerrit van der Laan, Mark C. Rosamond, Edmund H. Linfield, Gavin Burnell & Christopher H. Marrows Nature Communications volume 11, Article number: 428 (2020). Other works have suggested that the effect in the current driven systems arises due to the current changing the shape of the skyrmion. The role of temperature and drive current in skyrmion dynamics Kai Litzius, Jonathan Leliaert, Pedram Bassirian, Davi Rodrigues, Sascha Kromin, Ivan Lemesh, Jakub Zazvorka, Kyu-Joon Lee, Jeroen Mulkers, Nico Kerber, Daniel Heinze, Niklas Keil, Robert M. Reeve, Markus Weigand, Bartel Van Waeyenberge, Gisela Schütz, Karin Everschor-Sitte, Geoffrey S. D. Beach & Mathias Kläui Nature Electronics volume 3, pages30–36(2020) The results in the manuscript under review show the velocity dependence of the skyrmion Hall angle occurs even without an applied current, which suggests that current induced shape changes of the moving skyrmions are not key to producing the skyrmion Hall effect. The authors should mention the possible role of pinning on determining the Hall angel and for example if the authors see what appears to be a threshold, is this due to pinning or to some other effect?

We thank the Reviewer for mentioning these papers which discuss the velocity-dependence of the skyrmion Hall angle. The two theoretical papers by Reichhardt and co-authors considered a completely generic driving force that could also stem from a thermal gradient-induced magnon current. Therefore, the content of these papers also applies for our system and the velocity-dependence of the resulting skyrmion Hall angles qualitatively agrees with the present observation, see Fig.A1 (we have added Fig. A1 as Fig. S4 in the revised Supplementary Information). In principle, the thermal gradient and the resulting magnon current indeed also distort skyrmions in insulating systems. However, such significant effects as discussed by Litzius and co-authors are more likely in thin films where the skyrmions are large and soft, whereas skyrmions in our insulating chiral magnet Cu_2OSeO_3 are relatively rigid, see, e.g., J. Masell et al. Phys. Rev. B 101, 214428 (2020).

During the revision of the manuscript, we have corrected an error in the previous version of Fig. 3: The Hall angle cannot be defined when the skyrmion does not move at $v = 0$ in our present observations. Thus, we have removed the Hall angle data at $v = 0$ in the revised Fig. 3.

In summary, we agree with the Reviewer that our observations are nicely explained by pinning sites, i.e., both (i) the non-linearly increasing Hall angle as a function of the skyrmion velocity (Fig. A1) and (ii) the depinning threshold. We added the following discussions in line 44-line 53 of page 3 and line 173-line 179 of page 8-9 of the revised MS, including the above references.

“Recent research reveals the role of temperature in the dynamics of topological spin textures (skyrmions)¹³: the non-linearly increasing Hall effect as a function of the skyrmion velocity in the low-drive regime close to the depinning threshold; the current-induced the deformation of skyrmion

shape from a regular circle to an irregular one in the high-drive regime. A thermal gradient in metallic systems is also expected to cause skyrmion dynamics since the thermal current can induce both a magnon current and an electric current (thermopower). The electric current then exerts a force on the skyrmions via spin-transfer torque or spin-orbit torque, resulting in a complicated interaction between spin textures and heat current. These complex thermal effects have been demonstrated by a recent work on the temperature gradient-driven skyrmion diffusion in a metallic multi-layer film¹⁴.”

“Note that the Hall angle increases nonlinearly with the v , as shown in the Supplementary Figure 4. Qualitatively similar skyrmion Hall effects, i.e., the relationship between the Hall angle and velocity, have also been observed in experiments for electric-current-driven skyrmions^{13, 34-35} and were predicted theoretically for moving skyrmions in disordered media³⁶⁻³⁷. The velocity-dependence of the resulting skyrmion Hall angle shown here reveals a first experimental observation of heat-driven skyrmion Hall motion in the insulating magnet.”

Figure A1. The relationship of the skyrmion Hall angle and skyrmion velocity (v).

Response to Reviewer #2

The manuscript “Real-space observations of 60-nm skyrmion dynamics in an insulating magnet under low heat flow” presents the direct observation of translational and lateral motion of skyrmions driven by a temperature gradient in the insulating helimagnet Cu₂OSeO₃ using Lorentz transmission electron microscopy (Lorentz TEM). The temperature gradient is established by applying an electric current into the on-chip heater made of meander Pt stripes using electron lithography on the surface of Cu₂OSeO₃, and the 100 μ m-thick Cu₂OSeO₃ thin plate was partially thinned down to 100nm thick using a focused ion beam system so as to perform the in-situ Lorentz TEM observation. By changing the environment temperature T and normally applied magnetic field B , a phase diagram of magnetic structures in Cu₂OSeO₃ is established through analyzing the obtained Lorentz TEM images as well as the corresponding fast Fourier transforms (FFTs) results. Then the authors applied different electric current into the heater and estimated their effects on the

motion of skyrmions, during which a rather small current (10uA) can only change the orientation of skyrmions lattice, while obvious lateral and translational motion of skyrmion lattice can be witnessed when the current is higher (50uA and 100uA, respectively). The temperature gradient generated by electric current is quantitatively characterized using COMSOL simulation, and the temperature gradient needed in experiment for the velocity of skyrmion lattice to reach 0.1mm/s is of the same order with the theoretical estimation.

We thank the Reviewer for his/her evaluations of our present work.

1) The crystal structure, Figure 1A, the formation of skyrmions in this materials, Figure 1B, the phase diagram, Figure 2, for the magnetic structures of Cu₂OSeO₃ have already been published in many different articles [1–3]. The results produced by authors present subtle differences from those published results.

It is well known that the formation of skyrmions, lattice forms, and related phase diagrams depend on the external magnetic field, temperature, sample geometry (thickness, boundary condition), and other experimental procedures such as field cooling or zero-field cooling. The heat current-driven skyrmion dynamics are affected by the experimental conditions and we believe it is useful for the reader to provide these figures in the present work. Additionally, Refs. 1-3 are the literature for “spin caloritronics” and “spin Nernst effect” and are not articles regarding the magnetic structure in Cu₂OSeO₃.

2) Thus, this leaves only the schematic illustration in Figures 1c-1e, and Figure 3 as meaningful data in this manuscript. The discussion on these parts is, however, very limited. It is not clear how COMSOL simulation is being done, what are the parameters used for simulation?

We would thank the Reviewer for his/her question. We described our model and parameters for simulations using COMSOL in the “Methods” part in the main text. According to the Reviewer’s suggestion, we have expanded the model and all of the material parameters for COMSOL simulations in the revised Supplementary Information and as follows.

“Temperature map of finite element simulations

Supplementary Figure 3 shows the results of a finite element simulation performed using COMSOL Multiphysics commercial software. The model geometry is shown in Supplementary Figure 3a, with a ‘U’ shaped silicon base plate which is 0.3 mm thick and acts as the cold bath. The silicon substrate is contacted to the sample thin plate in two places: on the left-hand-side

using Ag paste (thermal conductivity $k_{iso_{Ag}} = 1 \text{ W}/(\text{m} \cdot \text{K})$, density $\rho_{Ag} = 1000 \text{ kg}/\text{m}^3$, and heat capacity $C_{p_{Ag}} = 50 \text{ J}/(\text{kg} \cdot \text{K})$), and on the right side using epoxy resin ($k_{iso_{epoxy}} = 0.2 \text{ W}/(\text{m} \cdot \text{K})$, $\rho_{epoxy} = 1100 \text{ kg}/\text{m}^3$, and $C_{p_{epoxy}} = 1110 \text{ J}/(\text{kg} \cdot \text{K})$). The Pt wire heater ($k_{iso_{Pt}} = 70 \text{ W}/(\text{m} \cdot \text{K})$, $\rho_{Pt} = 21447 \text{ kg}/\text{m}^3$, and $C_{p_{Pt}} = 130 \text{ J}/(\text{kg} \cdot \text{K})$) is located on the top right side of the Cu_2OSeO_3 thin plate ($k_{iso_{\text{Cu}_2\text{OSeO}_3}} = 40 \text{ W}/(\text{m} \cdot \text{K})$, $\rho_{\text{Cu}_2\text{OSeO}_3} = 5070 \text{ kg}/\text{m}^3$, and $C_{p_{\text{Cu}_2\text{OSeO}_3}} = 55.54 \text{ J}/(\text{kg} \cdot \text{K})$). The temperature gradient is calculated from these results to be $\nabla T \approx 11 \text{ mK}/\text{mm}$ for $I_H = 50 \mu\text{A}$ and $\nabla T \approx 4600 \text{ mK}/\text{mm}$ for $I_H = 1000 \mu\text{A}$.”

Supplementary Figure 3: Temperature maps of finite element simulations. **a.** Geometry of the device setup. **b-c.** Temperature map resulting from running an electric current (b) $I_H = 50 \mu\text{A}$ and (c) $I_H = 1000 \mu\text{A}$ through the heater wire. **d-e.** Line profiles extracted from the bulk (blue solid

line) and Lorentz TEM (green dashed line) regions of the sample marked (white dashed lines) in (b) and (c), respectively.

The identification of skyrmion motion, together with the device design are highly questionable. I am also wondering what is the purpose of mentioning 60 nm? Are there existing bigger skyrmions in Cu₂OSeO₃?

Manipulation of small-sized skyrmions is promising for future applications such as electronic devices with high-density information-carriers. We simply emphasize the tracking of heat-flow driven motion of such a small skyrmion, i.e., several tens nanometer scale, which would be difficult in conventional magneto-optical imaging adopted for the observation of skyrmionic bubbles of a few micrometer size. As is well known, the skyrmion size is proportional to the period of helical structure (λ), which is determined by the ratio of J/D (J and D are strengths of the exchange interaction and spin-orbital interaction, respectively) and depends less on the extrinsic parameters, such as B-field and temperature, etc. However, the size of skyrmions in chiral-lattice magnets can be tuned by an external magnetic field in the metastable state (X. Z. Yu, et al., *Nature Physics* 14, 832–836(2018)). In our present experimental setup, the skyrmion size is kept constant.

3) I am wondering why authors made their device out of a big thick plate? I am expecting to see more clean physics out of a single thin stripe, without complication from a big surrounding thick area.

The experimental design for our present work is beneficial for measuring the temperature gradient by measuring thermometer resistances settled on both cold and hot sides, as shown in Fig. A2. By contrary, it is challenging to realize in a free-standing lamella sample fabricated from a bulk piece due to the limited micrometer-scale sample size. In accord with the Reviewer's suggestion, we also prepared a lamella sample with a heater on the right side of the lamella. We then performed in-situ Lorentz TEM observations. The results are presented in Fig. A2. The skyrmions tend to flow from the cold side to the hot side, accompanied by the proliferation of skyrmions, in agreement with our present results. However, the temperature gradient needed to drive skyrmions in the lamella appears much larger than that for our current demonstration, possibly due to the random defects and boundary conditions in the confined geometrical lamella.

Figure A2 Skyrmion motions in a lamella with heat flows from right to left sides of a thin lamella sample. a. Scanning electron microscopy image of a lamella ($30\ \mu\text{m} \times 10\ \mu\text{m} \times 0.1\ \mu\text{m}$ in size) with a heater (W wire) on its surface, which is settled on a Si-membrane capped by a 50nm-SiN film. b. Schematic drawing of the W wire with the size of l (length) $\sim 30\ \mu\text{m}$, w (width) $\sim 0.3\ \mu\text{m}$ and t (thickness) $\sim 30\ \text{nm}$. The heater resistance is approximately $900\ \Omega$. c-d. Over-focus Lorentz TEM images observed (c) before and (d) during a heat flow (heater current is $300\ \mu\text{A}$) at 20 K with a normal field of 160 mT.

4) Under such a small temperature gradient, it is surprising to see the “motion” of skyrmions. While, I can see the propagation of skyrmions from left towards the right regimes, in the presence of temperature gradient. I failed to see a more comprehensive experimental investigation of the reversible skyrmion motion.

According to the Reviewer’s comments, we have shown the reversible skyrmion motion in another sample designed to flow heat current from left to right sides, as shown in Fig. A3. We create a

skyrmion cluster in the initial state (Fig. A3e). Accompanied by the proliferation of skyrmion, the cluster flows from the cold area (the right region in the Lorentz TEM image) to the hot (left) region with 100 μA -current flow (Fig. A3f).

Figure A3 Skyrmion motions in a thin Cu_2OSeO_3 with heat flows from left to right sides of the plate. a. Photo-microscopy image of a chiral-lattice insulator Cu_2OSeO_3 (2mm \times 1mm \times 0.1mm in size) with three Pt wires on its surface. H, R1 and R2 stand for the heater and thermometers, respectively. b. Schematic drawing of the Pt wire with the size of l (length) = 2.34 mm, w (width) = 2.1 μm and t (thickness) = 25 nm. c-d. Simulated temperature-contour maps in thin Cu_2OSeO_3 at heater-off (heater current $I_H = 0$; c) and heater-on (heater current $I_H = 100 \mu\text{A}$; d), respectively. e-f. Over-focus Lorentz TEM images observed (e) before and (f) with heat flow ($I_H = 100 \mu\text{A}$) at 20 K with a normal field of 45 mT.

The phase diagram suggests skyrmions can exist in a wide temperature range, I am thus wondering what limits the available amplitude of temperature gradients?

In principle, the temperature gradient can generate the heat current to drive skyrmion motion in the present chiral-lattice magnet as long as the sample temperature does not increase beyond T_C . Experimentally, we did not apply such large heater currents since our purpose was to observe the motion of skyrmions by a heat current flow. If the heat current is too large, the skyrmions move too fast and we cannot trace them anymore. We believe our result that very low heat currents can drive skyrmions is of astonishing novelty, while pushing this mechanism to the limits will be a necessary future work.

5) *I am not quite sure if it makes sense to talk about the skyrmion Hall angle in their experiment. After all, only skyrmions in particular region are movable. Their paths are thus confined/forced to that particular region (from bottom left to the up right), which in principle could give rise to a false estimation of skyrmion Hall effect, since their path is predetermined from the bottom left to the up right. Authors should reverse the current direction, and the polarity of magnetic field orientation, to see direction of skyrmion Hall motion.*

Skyrmions move in the opposite direction when the heat-flow direction is reversed. On the other hand, the Lorentz TEM view area is limited due to technical considerations (as we mentioned above, the larger-sized thin plate without supporting substrate is easily broken). Still, it should be large enough to present skyrmion dynamics with heat flows. We agree with the Reviewer's suggestion for texting skyrmion Hall motion with the flip of the external magnetic field. However, our Lorentz TEM provides only one directional field, and unfortunately, the field cannot be reversed for our in-situ Lorentz TEM observation. Concerning the skyrmion Hall effect, the quantitative analysis of the velocity-dependence of the Hall effect in Fig.A1 shows a substantial increase of the skyrmion Hall angle with increasing skyrmion velocity, which seems to converge to a finite value. This result agrees well with previous theories (see reply to Review#1) and indicates an *intrinsic* skyrmion Hall effect because *extrinsic* effects should vanish in the limit of large velocity.

6) *As I am concerned, the authors modeled the whole Cu₂OSeO₃ thin plate and perform the simulation using COMSOL in order to characterize the temperature profile of the sample, during which the very distant edge from the heater of the plate is set to be 20K directly, while the temperature elsewhere is calculated accordingly. I am not quite sure if this is a proper way to obtain the precise temperature, since the Cu₂OSeO₃ thin plate actually gets contacted with the "real" cold bath (the sample holder) through Si-substrate. It seems to me more convincing and reliable to simulate the whole device including Cu₂OSeO₃ thin plate and the Si-substrate, as the temperature profile is crucial for the subsequent analysis. Or can the author verify their simulation about temperature?*

We thank the Reviewer for his/her question related to simulations. We performed new simulations which include the Si substrate, as shown in the following, and appended detailed descriptions for COMSOL simulations in the revised Supplementary Information. The new simulations indicate the same order of ∇T but slightly smaller ∇T than the results shown in the main text.

Temperature maps of finite element simulations. **a.** Geometry of the device setup. **b,c.** Temperature map resulting from running an electric current (b) $I_H = 50 \mu A$ and (c) $I_H = 1000 \mu A$ through the heater wire. **d, e.** Line profiles extracted from the bulk (blue solid line) and Lorentz TEM (green dashed line) regions of the sample marked (white dashed lines) in (b) and (c), respectively.

7) Another major concern is the dragging of skyrmions back to the initial state after turning off the heater current is tuned off in Fig. 3c. What are the reasons underlying this observation? Does this mean that the sample is not uniform and the skyrmions are likely to stay in specific areas? Will the nonuniformity plays important roles in the motion of skyrmions as well (this question is similar to 5)

We thank the Reviewer for these constructive comments. Experimentally, skyrmions tend to return to the initial state when we stop the heat flowing through it, which was confirmed for three devices,

as shown in Fig. 1 as well as in Figs. A2-3. The mechanism behind this process is not clear at present; most likely, it is due to inhomogeneity, such as tiny residual strains in the sample, which determine the skyrmion distribution at the equilibrium state under zero heat low. When the heat current is switched off, the skyrmions can either move back to this energetically preferred position, or other skyrmions can be attracted by this spot, or new skyrmions can be nucleated. The precise mechanism is, however, not traceable within the time resolution of our experiments. We have added this discussion in the revised main text. A potential source of pinning areas where skyrmions preferably reside is thickness variations of the sample. We plot the thickness profile to the sample position in the thinner area for the view (Fig. A4). The thickness distribution shows an approximately constant thickness of 100 nm, yet some variations are visible. Such variation of the sample thickness, i.e., the surface roughness of the sample, may induce an inhomogeneous energy landscape for skyrmions, which influences the motion of skyrmions. For example, the critical thermal gradient for depinning skyrmions is likely to be a consequence of such disorder. Also, the velocity-dependence of the skyrmion Hall effect is influenced by disorder; see above discussion in reply to Reviewer #1. However, the impact of disorder vanishes in the limit of large velocities. Thus, we are convinced that the observed skyrmion Hall effect is indeed intrinsic.

Figure A4 Thickness profile with respect to the sample position in the view area.

8) In Fig. 3e-g, why does the left island marked with yellow circle move and unite with the right island rather than moving simultaneously with the right island, keeping the distance between them unchanged, if two islands can both be driven by the temperature gradient?

Yes, both islands can be driven by the temperature gradient but there is a delay or difference in moving these islands, indicating possibly different pinning potentials.

9) It should mention here that the interaction between skyrmions and heat can be very complex, and depend on the detailed parameters, many different physics and direction of motion can be observed. For examples, a motion of skyrmions from hot towards cold regimes in metallic multilayer has been reported. This suggests a more comprehensive theoretical discussion, based on the available experimental parameters in their material system should be adequately discussed.

We thank the Reviewer for this suggestion. We add the following discussion and related reference at the end of the first paragraph of the revised MS.

“Recent research reveals the role of temperature in the dynamics of topological spin textures (skyrmions)¹³: the non-linearly increasing Hall effect as a function of the skyrmion velocity in the low-drive regime close to the depinning threshold; the current-induced the deformation of skyrmion shape from a regular circle to an irregular one in the high-drive regime. A thermal gradient in metallic systems is also expected to cause skyrmion dynamics since the thermal current can induce both a magnon current and an electric current (thermopower). The electric current then exerts a force on the skyrmions via spin-transfer torque or spin-orbit torque, resulting in a complicated interaction between spin textures and heat current. These complex thermal effects have been demonstrated by a recent work on the temperature gradient-driven skyrmion diffusion in a metallic multi-layer film¹⁴.”

References:

- Kai Litzius, et al. The role of temperature and drive current in skyrmion dynamics. *Nat. Electronics* **3**, 30–36(2020)
- Zidong Wang, et al. Thermal generation, manipulation and thermoelectric detection of skyrmions. *Nat. Electronics* **3**, 672–679 (2020)

In metallic systems, the situation is therefore quite complex. However, since we study an insulator, these electronic effects are absent and the main mechanism is expected to be the magnon current, see Refs. 23,32-33. In particular, Ref.33 points out the differences between metallic and insulating systems, which nicely explains the discrepancy between the previous work by Wang et al. and our work.

10) Many early (mostly theoretical) discussions on the interaction between skyrmions and heat did not get noticed. The cited references on spin caloritronics are too general and not specific for the present manuscript.

We thank the Reviewer for the suggestion. There are indeed many papers which study skyrmions and heat, but usually the temperature is uniform. We decided to only include the most relevant papers which study skyrmions in temperature gradients. If we overlooked important and relevant papers during our intensive literature research, we would of course be most grateful if the Reviewer could supply more precise information.

To improve our manuscript, we have added several references as follows in the revised MS.

- Zidong Wang, et al. Thermal generation, manipulation and thermoelectric detection of skyrmions. *Nat. Electronics* **3**, 672–679 (2020)
- Reichhardt, C., Ray, D. & Reichhardt, C.J. Collective transport properties of driven Skyrmions with random disorder. *Phys. Rev. Lett.* **114**, 217202 (2015).
- Reichhardt, C. & Reichhardt, C.J. Noise fluctuations and drive dependence on the skyrmion Hall effect in disordered systems. *New J. Phys* **18**, 095005 (2016)
- Katharina Zeissler, et al. Diameter-independent skyrmion Hall angle observed in chiral magnetic multilayers. *Nat. Commun.* **11**, 428 (2020)
- Wanjun Jiang, et al., Direct observation of the skyrmion Hall effect. *Nat. Phys.* **13**, 162 (2017)
- Kai Litzius, et al. The role of temperature and drive current in skyrmion dynamics. *Nat. Electronics* **3**, 30–36(2020)

11) Authors observed the motion of skyrmion from cold towards hot regime. Their experimental observations demand further comprehensive investigations.

We thank the Reviewer for the comments. We demonstrated skyrmion motion in two samples by reversing the heat flow direction. It is discerned that the skyrmions move from the cold to the hot sample areas in both devices. Experimental results demonstrated here show a clear evidence for skyrmion Hall motions driven by heat flows, which are in agreement with theoretical predictions (Refs. 23, 29-33, 36-37).

Whether the motion of skyrmion is a result of magnonic spin torque remains further justification.

In this insulating system, the scenario that the magnonic current arising from the temperature gradient can be a main driving force for skyrmion dynamics has already been confirmed by previous theoretical studies (Refs. 23, 30-33), which are all in accord with the present observations. We

cannot imagine some other plausible mechanisms to account for the heat-current-driven skyrmion motions.

The observation of skyrmion Hall angle is not adequately discussed, which probably occurs as a result of the uniformity of their sample. The theoretical discussion is rather vague, the simulation requires a more careful revisit. This manuscript requires a substantial experimental input.

We thank the Reviewer for his/her critical comments. As shown in Fig. A4, we agree with the Reviewer's criticism; the sample exhibits surface roughness or random distribution of impurities caused by Ga-ion sputtering which was used to thin the bulk sample. Such surface roughness or random distribution of contaminants can affect the skyrmion Hall motion. However, the impact of impurities vanishes in the limit of large skyrmion velocities. Also, the relationship of skyrmion Hall angle and velocity observed here quantitatively agrees with previous experimental observations for skyrmion Hall motions with electric current driving (Wanjun Jiang, et al., Direct observation of the skyrmion Hall effect. Nat. Phys. 13, 162 (2017); Kai Litzius, et al. The role of temperature and drive current in skyrmion dynamics. Nat. Electronics 3, 30–36(2020)), and theoretical predictions for skyrmion dynamics in the randomly disordered system (Reichhardt, C. & Reichhardt, C.J. Noise fluctuations and drive dependence on the skyrmion Hall effect in disordered systems. New J. Phys 18, 095005 (2016)). Thus, we are convinced that the observed skyrmion Hall effect is indeed intrinsic.

All in all, we have responded one-by-one to Reviewer#2's comments and questions as mentioned above, although not all the issues could be fully resolved at the moment in the present study from the Reviewer#2's critical viewpoints. The heat current that drives skyrmion dynamics is vital for fundamental topological physics and future electronic devices. The demonstration of dynamics of such minute topological objects at liquid helium temperatures is challenging work, for which we've spent five years. Because of this difficulty, experimental studies on this phenomenon are rare. Our present work demonstrates skyrmion dynamics with such small heat flow, two orders of magnitude smaller than that which drives ferromagnetic domain walls, as Reviewer#1 recognized. Our in-situ real-space observations reveal skyrmion Hall motion driven by the temperature gradient. However, the scenario for "*the dragging of skyrmions back to the initial state after turning off the heater current is turned off*" is not clear. We believe this experimental work should inspire researchers for *further studies on skyrmion dynamics in the single and lattice limit under thermal gradients*, as Reviewer#1 emphasized.

Reviewers' Comments:

Reviewer #1:

Remarks to the Author:

The authors have made some very nice changes to address the points I raised in the first round and I am now satisfied. The authors also made several changes to address a different set of points raised by the second referee and these changes also improve the paper in opinion. Overall I think the observation of thermal gradient driven motion on a collection of skyrmions and the changes in the Hall angle is very interesting and will be an important result in the field. I now recommend the paper for publication now.

Reviewer #2:

Remarks to the Author:

Authors have addressed most of my questions and comments satisfactorily. While I can see the experimental data and explanation presented in the manuscript appears to be incomplete, the key message of the manuscript is clearly delivered. Namely, the motion of skyrmions driven by (reversed) thermal gradients in insulating materials, which is partly supported by experimental and theoretical efforts. Considering the current development in the skyrmion research community, the challenge in performing additional experiments, I can suggest its publication after addressing the following aspects. However, I must emphasize here that:

1: The motion of skyrmion driven by such a small thermal currents, should be in the creep motion regime, the discussion of skyrmion Hall angle may not be relevant (which occurs in the flow motion regime).

2: Even if there is a skyrmion Hall effect, in the lattice, one should also observe a lattice rotation. Similar gyromotion of skyrmions driven by random thermal fluctuation has been reported.

3: The magnonic current is lack of quantitative/in-depth discussions. Based on the temperature gradient, one can estimate the amplitude of such current. After all, it is not the temperature gradient, but the magnonic currents and magnetostatic gradient induced by the temperature gradients that produce the subsequent motion.

I feel that these three points should be further improved before it can be published.

Response to Reviewers' Comments

The following shows our point-by-point responses to Reviewer's comments. For better visibility, we cite the reviewers' comments in *blue*.

Response to Reviewer #1

The authors have made some very nice changes to address the points I raised in the first round and I am now satisfied. The authors also made several changes to address a different set of points raised by the second referee and these changes also improve the paper in opinion. Over all I think the observation of thermal gradient driven motion on a collection of skyrmions and the changes in the Hall angle is very interesting and will be an important result in the field. I now recommend the paper for publication now.

We thank the Reviewer for reviewing our revised manuscript. We are grateful to his/her recommendation to publish our revised manuscript in Nature Communications.

Response to Reviewer #2

Authors have addressed most of my questions and comments satisfactorily. While I can see the experimental data and explanation presented in the manuscript appears to be incomplete, the key message of the manuscript is clearly delivered. Namely, the motion of skyrmions driven by (reversed) thermal gradients in insulating materials, which is partly supported by experimental and theoretical efforts. Considering the current development in the skyrmion research community, the challenge in performing additional experiments, I can suggest its publication after addressing the following aspects.

We thank the Reviewer for reviewing our revised manuscript. We are grateful to the Reviewer's evaluation for our revised manuscript and our challenging work.

1: The motion of skyrmion driven by such a small thermal current, should be in the creep motion regime, the discussion of skyrmion Hall angle may not be relevant (which occurs in the flow motion regime).

We thank the Reviewer for the comment on the skyrmion motion observed in the present

work. As the Reviewer points out, the skyrmion motion under such a small temperature gradient invokes a possibility of the creep motion. Before discussing our experimental results, let us refer to the numerical simulation (ref. 37). As shown in Fig. e, the creep motion is characterized by inhomogeneous and incoherent skyrmion motion, and the intrinsic Hall angle can therefore be discussed only when sufficient statistical averaging is performed; otherwise, the observed Hall angle is merely a reflection of a local profile of the impurity potential. We think the Reviewer is concerned about this point. In contrast, in the flow regime, the skyrmion motion is homogeneous and coherent, and one can therefore discuss the intrinsic Hall angle from arbitrary position.

When having a look at our LTEM observation of the skyrmion motion (Figs. a-b) at a low temperature gradient (marked by a blue arrow in Fig. g) corresponding to a heater current of $10 \mu\text{A}$, the rotating motion of skyrmions occurs, but the Hall motion is suppressed. When the temperature gradient increases to a critical value related to a heater current of $50 \mu\text{A}$ (marked by a black arrow in Fig. g), one can notice that the skyrmions move coherently over a length of ~ 100 skyrmion lattice constants (the lower part of Fig. c,d). We think that such coherence is a hallmark of the skyrmion flow and therefore the Hall angle reflects the intrinsic behavior, rather than something dominated by local profiles of the impurity potential. Furthermore, as the temperature gradient increases, the Hall angle smoothly increases and saturates (Fig. g). But we should note that all skyrmions do not show such a flow motion; for instance, please see the upper part of the Fig.c,d, where the skyrmions do not move clearly. This observation implies that the pinning strength is inhomogeneous in real materials. Our Hall-angle analysis is performed only for the skyrmion motion with coherence.

Figure: a-d, g extracted from Fig. 3 in the main text, e-f reproduced from ref. 37, respectively.

a-d show Lorentz TEM images observed before (a, c) and during (b, d) $10\text{-}\mu\text{A}$ (b) and $50\text{-}\mu\text{A}$

(d) current flowing through the heater, respectively. e-f show simulated creep (e) and flow (f) motions at driving forces of 0.02 and 0.2, respectively. g shows a profile of Hall angle (blue triangular) of the front skyrmion (marked by cups dashed lines in c and d) with an increase of ∇T . The pink and blue solid lines are eye guides for the changes of the skyrmion velocity and Hall angle, respectively. The blue and black arrows indicate data points observed at the heater currents being 10uA and 50uA, respectively.

On the other hand, we have corrected the skyrmion velocity in the revised manuscript, which is also shown in Fig. g. The corrected velocity value is considerably smaller than the previously described values. The difference is caused by two reasons as follows. 1) the skyrmion velocity is not constant during the heater current applications (see details in the revised supplementary information), and only fast motions were presented in the previous manuscript. In the revised manuscript, we plotted the averaged velocity (\bar{v} in Fig. g) as the ratio of the total drift distance to the duration (the duration is 1.36 s for the heater current $I_H = 0.3$ mA, 0.5 mA, while it is 1.68 s for $I_H = 0.3$ mA, 0.05 mA, 0.1 mA) of skyrmion motion. The error bars represent the maximal and minimal velocities deduced from the locally-averaged values approximately over the shorter time period of 0.2-0.6 s, as detailed in the newly added Supplementary Figure 1. 2) We checked the raw data of Lorentz TEM movies and found a mistake in the exposure time to record Lorentz TEM images. The correct exposure time is 40 ms, not 5 ms which was erroneously used in the previous manuscript. We apologize for this serious mistake to cause the possible confusion. However, our conclusion for skyrmion dynamics induced by the temperature gradient, including the critical temperature-gradient value, is not affected by this revision.

2: Even if there is a skyrmion Hall effect, in the lattice, one should also observe a lattice rotation. Similar gyromotion of skyrmions driven by random thermal fluctuation has been reported.

We thank the Reviewer for the comment on the rotating motion of skyrmion lattice. As we presented in Figs. 3a-3b, the orientation of skyrmion-cluster changes with a relatively small temperature gradient, meaning that the rotating motion occurs, but the Hall motion is suppressed. The Hall motion of the collective skyrmion state shows up under a relatively larger temperature gradient, as shown in Figs. 3c-3d, wherein the rotating motion of the collective skyrmion state is suppressed. The driving force for the present skyrmion Hall motion is thermal current, which is distinct from random thermal fluctuation and geostrophic pinning force. Therefore, the skyrmion Hall motion dominates over the thermal-fluctuation

induced gyromotion, in our temperature gradient-driven skyrmion dynamics.

3: The magnonic current is lack of quantitative/in-depth discussions. Based on the temperature gradient, one can estimate the amplitude of such current. After all, it is not the temperature gradient, but the magnonic currents and magnetostatic gradient induced by the temperature gradients that produce the subsequent motion.

We thank the Reviewer for the constructive comment and suggestion for quantitatively analyzing the magnonic current. The scope of the present work is to demonstrate the thermal current-driven skyrmion motion as well as the relationship among the skyrmion velocity, Hall angle, and temperature gradients, while the thermal current should certainly lead to the magnonic current as a possible microscopic carrier of the skyrmion. In the previous versions of the manuscript, we have argued that the motion of skyrmions is induced by the magnon current and have presented the corresponding theoretically calculated skyrmion velocity. This calculation has previously been discussed in various theory papers (Refs. 14, 31, 32), therefore we think that our manuscript would not benefit from a detailed technical repetition of these already published results. However, in response to the Reviewer's comment, we added the following sentence in lines 191-192 of revised MS:

"... we roughly estimate a magnon current of the order $j_x \sim -9 \times 10^{20} \nabla T$ ($\text{K}^{-1} \text{s}^{-1} \text{m}^{-1}$) skyrmions (with ∇T in the unit of K/m) which results in magnon-induced skyrmion velocity".

While checking our calculations for the magnon current and skyrmion velocity again, we realized that the previously assumed Gilbert damping constant was too small ($\alpha = 10^{-3}$). This was due to a typo in the cited reference and should have been $\alpha = 6 \times 10^{-3}$ instead. The corrected (theoretically predicted) skyrmion velocity is $8 \mu\text{m}/\text{s}$ at a temperature gradient of $50\text{mK}/\text{mm}$. We apologize for this mistake.

From the viewpoint of experimental proof, the quantitative evaluation of magnonic current is an important future task.

Reviewers' Comments:

Reviewer #2:

Remarks to the Author:

Authors clarified necessary parts associated with the manuscript. I suggest its publication in Nature Communications.